# Using Organotypic Tissue Slices to Investigate the Microenvironment of Pancreatic Cancer: Pharmacotyping and Beyond

**DOI:** 10.3390/cancers13194991

**Published:** 2021-10-05

**Authors:** Jonathan Robert Weitz, Herve Tiriac, Tatiana Hurtado de Mendoza, Alexis Wascher, Andrew M. Lowy

**Affiliations:** 1Department of Surgery, Division of Surgical Oncology, Moores Cancer Center, University of California, San Diego, La Jolla, CA 92037, USA; jweitz@health.ucsd.edu (J.R.W.); htiriac@health.ucsd.edu (H.T.); tchurtadodemendoza@health.ucsd.edu (T.H.d.M.); awascher@ucsd.edu (A.W.); 2Moores Cancer Center, University of California, San Diego, La Jolla, CA 92037, USA

**Keywords:** pancreatic cancer, PDAC, microenvironment, organotypic, slices, 3D culture

## Abstract

**Simple Summary:**

Pancreatic ductal adenocarcinoma (PDAC) has the highest mortality rate of all major cancers and, disappointingly, neither immune- nor stroma-directed therapies are found to improve upon the current standard of care. Among the most challenging aspects of PDAC biology which impede clinical success are the physiological features of the pancreatic cancer microenvironment (TME), including the presence of a highly fibrotic extracellular matrix marked by perineural invasion and an immunosuppressive milieu. Many current strategies for PDAC therapy are focused on altering these features to improve therapeutic efficacy. This review discusses the recent investigations using organotypic tumor slices as a model system to study cellular and extracellular interactions of the pancreatic TME. Future studies utilizing such models may provide new insights into the TME by identifying mechanisms of communication between multiple cell types and investigating novel therapeutic approaches for personalized cancer therapy.

**Abstract:**

Organotypic tissue slices prepared from patient tumors are a semi-intact ex vivo preparation that recapitulates many aspects of the tumor microenvironment (TME). While connections to the vasculature and nervous system are severed, the integral functional elements of the tumor remain intact for many days during the slice culture. During this window of time, the slice platforms offer a suite of molecular, biomechanical and functional tools to investigate PDAC biology. In this review, we first briefly discuss the development of pancreatic tissue slices as a model system. Next, we touch upon using slices as an orthogonal approach to study the TME as compared to other established 3D models, such as organoids. Distinct from most other models, the pancreatic slices contain autologous immune and other stromal cells. Taking advantage of the existing immune cells within the slices, we will discuss the breakthrough studies which investigate the immune compartment in the pancreas slices. These studies will provide an important framework for future investigations seeking to exploit or reprogram the TME for cancer therapy.

## 1. Introduction

The role of the tumor microenvironment (TME) has come to the forefront of cancer research since the ideological conception of the “seed and soil” theory by Stephen Paget in 1889 [1]. As the analogous “seed”, cancer cells are programmed via genetic and epigenetic alterations which encode for a replicative potential to create a malignant neo-organ. However, without an accommodating niche or “soil”, these cancer cells fail to persist. Incapable of self-sustenance, cancer cells shape their local microenvironment to provide the basic necessities for their survival: (1) growth factors to support cell proliferation, (2) a vasculature for the delivery of oxygen and nutrients, (3) stromal matrix deposition for a stable cytoarchitecture, and (4) shelter from immune surveillance. These requirements to maintain cancer cell survival are supplied by multiple cell types including endothelial, fibroblast, nerve, and immune cells. The coalescence of these “normal” cells, alongside cancer cells, form a functionally growing tumor. As a result, destabilizing one or more components of the tumor microenvironment has become a major focus of investigation for anti-cancer therapies.

Pancreatic ductal adenocarcinoma (PDAC) is a particularly lethal cancer due to late diagnosis, early metastasis and therapy resistance [2]. While mouse models such as the KrasLSL-G12D/+; Pdx1-Cre [3] and KrasLSL-G12D/+; Trp53LSL-R172H/+; and Pdx1-Cre [4] were particularly useful for studying several aspects of pancreatic cancer biology, thus far therapies evaluated primarily in these preclinical models were not translated into improved clinical outcomes for patients. As a result, the additional models which recapitulate human pathophysiology and disease progression are needed. To this end, ex vivo models including tumor organoids and organotypic slices were developed from human specimens during the last ten years [5,6,7,8]. While these models offer unique approaches to the study aspects of pancreatic tumor biology, each have their advantages and shortcomings. As the functions of the tumor microenvironment are of increasing interest, organotypic tissue slices are well-suited to address the many unresolved questions in the field of PDAC biology. 

## 2. History and Origins of Pancreatic Slice Culture

The early pioneering studies using pancreas organotypic slice cultures characterized endocrine and exocrine cell physiology from mouse [9] and human tissues [10]. In addition to endocrine and exocrine cell types, the ex vivo preparations of precision cut slices (150–350 microns thick) from healthy donors also showed that immune [11,12] and vascular [13] cell populations were found to be maintained during slice culture. While severed from blood flow, cells within the tissue slices maintained their functional activity including hormone secretion [14,15], vascular contractility [13], cytokine secretion, and calcium signaling [11,12]. During the last decade, the use of slices to characterize cellular function during normal physiology has become an important tool for understanding the etiology of pancreatic diseases, such as type 1 diabetes [14] and cancer [16]. Further studies on disease modeling using pancreatic tissue slices should be carefully designed to take advantage of this ex vivo platform containing multiple biological components.

## 3. Tumor Culture Models as Tools for the Investigation of Multiple Cellular Component Systems: Synthetic Approaches, Organoids, and Organotypic Slices

Since the inception of the initial cell culture experiments in the 19th century, multiple platforms were established to investigate the biology of individual cells, tissues, and organs. Cell culture systems are utilized to maintain cellular and tissue integrity outside of a native organism, in vitro. As the elemental building blocks, individual cells require the most basic components for survival, including controlled temperature and gas levels, osmolarity, pH, substrate for cell attachment and nutrients. Once assembled into tissues, conditions allowing for the maintenance of cellular homeostasis become exponentially more complex. Here, we discuss the current approaches to study cellular and multicellular interactions in the pancreatic cancer microenvironment, highlighting recent discoveries, as well as fundamental differences in the culture methods and platform functionality between synthetic cellular approaches, organoids and organotypic tumor models.

### 3.1. Synthetic Approaches for 3D Reconstruction of the PDAC Microenvironment; Organoid Co-Cultures, Hydrogels, and Microfluidics Devices

Consisting of epithelial cells aggregated as 3 dimensional spheroids, organoids, or encapsulated cells in scaffold-based hydrogels, these methods use a synthetic approach, in which, a priori cellular components are re-assembled to reconstruct a biological system. In PDAC organoids, epithelial cells are grown as 3D self-assembling structures from murine [17] or human tumors [6,8]. In addition to epithelial cell cultures, to determine paracrine signaling communication networks, co-culture experiments are performed using fibroblasts [18,19] infiltrating lymphocytes [18,20], as well as nerves [21]. While these experiments can be continually propagated and are easily repeatable, the cytoarchitectural arrangement of the cells remains an approximation of the in vivo setting. As such, systems using microfluidic “organ-on-a-chip” systems have recently been developed as a potentially useful intermediate experimental platform to allow for the incorporation of multiple cellular components, vascular-like fluidics, and an ECM bio-scaffold. However, these devices are often independently developed and not always available for consumer purchase. As with all in vitro studies, to confirm if experimental findings are representative of in vivo biology, a subsequent validation in vivo is required.

Monolayer cell lines are used extensively to test therapeutics, but these studies often fail to translate into clinical benefits [22]. The 3D culture systems such as organoid cultures show promise. Multiple groups isolated large cohorts of patient-derived organoids from PDAC patients using surgical or biopsy material and tested rational therapeutic strategies [6,23,24,25]. While some of these studies demonstrated correlations between organoid and patient outcomes when matching therapies were tested, these organoid cultures lacked a stromal compartment; therefore, immunotherapy and other stromal targeting strategies could not be evaluated. Organoid co-cultures enabled the study of cell–cell interactions and facilitated the testing of stromal-targeted therapeutics; however, these methods relied upon human cells that were typically not autologous or syngeneic [26,27]. Using an air liquid interface (ALI) culturing method, Neal and colleagues demonstrated that freshly isolated patient-derived organoids could maintain both fibroblast and immune compartments [28]. In a proof of concept experiment, the authors were able to test anti-PD1 immunotherapy in ALI organoid cultures within a short time frame. More recently, Kokkinos et al. explored the capability of PDAC explants to recapitulate the therapeutic outcomes of a patient [29]. A surgically resected tumor tissue was cut into thick sections that were cultured for 12 days. These explants maintained stromal compartments mirroring the composition of the stromal cells and ECM observed in the original tumor. Encouragingly, the authors demonstrated a differential activity in the standard of chemotherapy care in a proof-of-concept experiment. This explant platform offered a highly reliable method for culture, transfection, drug testing and molecular assays. However, due to the high opacity caused by thick tissues, the live cell imaging of explants (1–2 mm) is more challenging compared to slice cultures (150–350 microns).

Patient-derived organoids, ALI and explant cultures are complimentary methods that recently empowered researchers to test therapeutic strategies in vitro rapidly and efficiently. Using these approaches, the standard of care therapies can be tested to potentially impact clinical decisions; however, the ultimate evaluation of matched cultures and patient outcomes must be performed in the setting of a clinical trial. The predictive power of organoid monocultures isolated from advanced pancreatic cancer patients is currently being tested in a randomized Phase II trial (NCT04469556). If therapeutic responses in organoids demonstrate a significant correlation with the clinical response, they will offer scientists and clinicians a robust platform to rapidly evaluate cancer-directed therapies. Alternative systems such as organoid co-cultures, ALI and explants will be needed to empirically test stromal-targeted therapies and these models will also require an evaluation in well-controlled clinical trials.

### 3.2. Ex Vivo Approaches Using Primary Tumor Tissues; Organotypic Slice Cultures

When in vivo models are not readily available or practical, patient-derived primary tumor slices are a valuable resource for researchers that allow for the functional interrogation of a tumor ex vivo. The preparation and maintenance of tumor slices in cultures begins with the removal of a viable tumor from a patient, generally via surgical resection, though slices can also be produced from core needle biopsy. To prepare the tumor sample for slicing using a vibratome, the isolated tumor is often trimmed of biomaterial which is extraordinarily collagenous, mucinous, or “sticky”, since it interferes with the physical cutting of the oscillating blade of the vibratome. After trimming, the tumor is embedded in an agarose gel and mounted on a vibratome stage for precision cutting. If the tumor is improperly trimmed, a failure in slice production will occur due to the tissue becoming stuck to the blade, causing a dislodging of the tissue from the agarose. When deciding to cut a desired thickness of a precision cut slice (between 150–350 microns) it is important to consider downstream analyses, since live-cell imaging is more challenging using thicker tissues [10,30,31]. Upon the successful sectioning of the tumor, the slices are cultured on cell-culture inserts, which allow for the uniform distribution of oxygen and increased slice viability [15]. It is important to maintain caution when handling the slices so that they do not stack or fold, which can result in reduced oxygenation. Given that multiple cell types, such as stromal, immune, and epithelial cells are present within organotypic slices, ex vivo PDACs are cultured in basal media that is supportive of these various cell types [30,31]. Multiple reports confirmed that immune cells remained viable during extended slice cultures using basal media (DMEM or RPMI1640 + 10% FBS) [16,31,32]. Additionally, the investigations of protein stability and function from human PDAC tissue slices revealed that proteins which were required for immunological function remained largely intact during the extended slice culture [16]. These results suggesedt that the endogenous signaling from local cells was sufficient to maintain leukocyte viability, as well as functional secretion of cytokines [32]. Given that the immunological function remains intact, future studies modulating adaptive and innate immune cells should continue to leverage this platform as a pre-clinical platform for the investigation of novel immunotherapies.

While basal media may be sufficient for maintaining the viability of some cell types during organotypic culture, other specialized supplements are often needed for the health and integrity of more fragile cell types. For instance, sympathetic and sensory neurons require a nerve growth factor (NGF) for survival, proliferation, and development [33,34,35]. As the endogenous release of NGF, as well as other growth factors may be present in tissue slice culture (i.e., released from neighboring cells), these factors remain undefined and are essential to characterize in future experiments investigating neuro–tumor crosstalk. This does not preclude studies using slices under acute culture conditions, given that nerve activity remains functional in the short-term cultured healthy pancreatic tissue slices [36]. Studies comparing the differences between healthy pancreatic nerves and pathological perineural invasion using slice culture will provide context to meta-analyses findings which show that perineural invasion contributes to a poor clinical outcome and the recurrence of pancreatic cancer [37,38].

## 4. Tumor Slices as a Model to Study Cellular and Acellular Interactions in the TME

The PDAC tumor microenvironment plays a crucial role in cancer growth and metastasis, drug delivery and resistance to therapy [2]. It is composed of cancer-associated fibroblasts (CAFs), immune and vascular cells immersed in an extracellular matrix that contains collagen, fibronectin, laminin, hyaluronic acid, proteoglycans, proteases, and other enzymes [39]. In this section we discuss the recent discoveries using slice culture to investigate paracrine communication networks in the cellular and acellular compartments of PDAC. 

### 4.1. Paracrine Signaling Interactions in the Cellular Compartment during PDAC

Several groups showed that PDAC slices from surgical specimens maintained their cellular integrity and composition. The immunostaining with markers of different cell populations, such as EPCAM for epithelial cells, αSMA for CAFs and CD3, CD20, and CD68 for T cells, B cells, and macrophages, respectively showed that these aforementioned cellular populations were preserved after more than one week in culture. Ki67 staining confirmed that there was an active proliferation ex vivo [16,31,32]. In addition, an analysis of the mTOR pathway showed that the slices were metabolically active [31]. Furthermore, the incubation with CFSE-labeled autologous immune cells from the spleen or peripheral blood showed the incorporation of these cells into the slices, providing a useful tool to study adoptive cell transfer therapies [16]. 

One of the best examples of research investigating cellular interactions within the TME in the context of tumor slices comes from Seo et al. These authors studied the distribution of immune cells within PDAC. They observed that stroma regions adjacent to the epithelial cancer cells contained fewer CD8 T cells, but more regulatory T cells (T regs) and tumor-associated macrophages (TAMs). In the same study, this unique arrangement of immune cells was also found in lymphatic tissues [32]. These findings suggested that the cancer cells interacted with the surrounding stroma to block the infiltration of cytotoxic T cells and attract immunosuppressive cells, such as T regs and TAMs.

The authors observed a higher PD-1 expression in CD8+ T cells in stroma without neighboring cancer cells. Using organotypic tumor slices, they tested the effect of CXCR4, and the PD-1 blockade, either alone or in combination, by treatment with AMD3100, a small molecule that inhibits CXCR4 and a PD-1 blocking monoclonal antibody, a strategy previously described [40]. The PDAC slices treated with the combination therapy were marked by a greater fraction of caspase-positive, apoptotic cancer cells compared to untreated, or monotherapy-treated slices. More importantly, there was an increased infiltration of both CD4 and CD8 T cells in the combination or anti PD-1 therapy groups, due to activation and expansion of the tumor-resident T cells, as seen by the increased granzyme and interferon gamma production. Taking advantage of the intact tumor cyto-architecture, the authors performed a time-lapse, confocal microscopy and were able to observe how T cells migrated to the epithelial cancer cells from regions of stroma devoid of these cells. Following migration, an increased apoptosis was observed in the EPCAM-positive cancer cells. The therapeutic benefit was observed exclusively in the combination therapy-treated slices, reiterating the importance of addressing not only obstacles in the immune compartment (PD-1), but also their interactions with the CAFs (CXCR4/CXCL12 axis) [32]. In this case, the organotypic tumor slice platform provided a unique opportunity to observe the interaction between tumor cells, CAFs and immune cells in real time and in their original spatial configurations, making this an ideal system to test new therapeutic strategies for PDAC. Such methods are clearly valuable in the studies of paracrine signaling and may be further enhanced by methods previously established using healthy pancreas tissue slices, such as live cell calcium imaging, to define novel paracrine signaling interactions in the tumor microenvironment (Figure 1).

### 4.2. Acellular Biophysical Interactions in the Tumor Slice Microenvironment

The acellular cytoarchitectural structure is a critical component of the PDAC microenvironment that contributes to tumor metastasis [41], the poor perfusion of anticancer therapeutics [42], low vascularity [43], and an increased interstitial pressure and stiffness [44,45]. In addition to the physical features of the extracellular matrix (ECM), which contribute to cancer cell survival, cells can scavenge the extracellular matrix proteins, such as albumin and proline, under nutrient stress conditions [46,47]. The inhibition of pathways regulating the scavenging of proteins and amino acids via macropinocytosis impairs the proliferation in PDAC cells [48]. Given this therapeutic potential, numerous clinical trials to target the PDAC ECM were conducted and others are in progress, as reviewed in [49]. Thus far, these trials have not resulted in improved outcomes for PDAC patients. While there are many possible explanations for the results, it is clear that there are undefined complex paracrine interactions between the acellular and cellular components of the PDAC tumor which remain critical to address. Using pancreatic tissue slices may ultimately serve as a critical tool needed to help resolve these unanswered questions. 

As a critical step to determine the ECM features of PDAC, it is important to apply the available technologies to the assay functional properties of ECM. These technologies include the topographical structure, determination of the acellular content (proteins and proteoglycans), as well as the stiffness and elasticity. Over the last decade, non-labeling strategies to image the topographical structure were performed in healthy living pancreas tissues by measuring tissue backscatter. To measure backscatter, an incident laser beam at an angular position of 180 degrees for imaging on a confocal microscope was used in order to visualize features of pancreatic tissue topography including: cellular volume, granularity, and matrix structure [10,50]. It will be important to utilize these tools to fully visualize spatial relationships between cellular and extracellular structures in PDAC slices. 

To expand upon the role of ECM in PDAC biology, the slice cultures offered a method for the functional interrogation of the biological processes regulating ECM turnover (secretion and breakdown). In a recent study by Li et al., the authors investigated the role of hypoxia on the secretion of the proteoglycan and lumican from pancreatic stellate cells. As an inhibitor of cancer progression, it is important to understand how its secretion and expression by stellate cells is regulated. The authors showed that lumican secretion was significantly reduced from tissue slices under hypoxic culture conditions compared to a normoxic culture [51]. Ultimately, the integration of ECM biology into a working model to test strategies which prevent, eradicate, or immunologically target cancer cells is the goal of such research. As an example of this, a pioneering study from Boluda et al., investigated how tumor stiffness acted as a physical barrier for T cells, thus preventing the T cell targeting of tumors using checkpoint inhibitors. Specifically, the authors used fresh tumor slices from mice treated with beta-aminopropionitrile (BAPN), a lysyl oxidase (LOX) inhibitor regulating collagen fiber stabilization. The authors showed that BAPN interfered with collagen stabilization in tumor-bearing KPC mice, reduced the ECM content, tumor stiffness, improved T cell migration, and increased efficacy during the anti-PD-1 blockade [52]. As such, this study highlights the potential clinical relevance of integrating immunotherapy with ECM therapy, while using tissue slices as an investigational tool for personalized medicine (Figure 2).

## 5. Challenges and Limitations of PDAC Slice Cultures

Organotypic slices offer a unique approach to study the tissue microenvironment in a semi-intact preparation; however, the suboptimal culture conditions resulting in cell death may produce unintended experimental biases. For example, acinar cells within the exocrine pancreas contain enzymes such as trypsin, amylase and lipase that aid in food digestion. These can negatively impact the survival of pancreatic slice cultures if these enzymes are released and activated in the culture supernatant. Investigators using PDAC slices cultures containing acinar cells, should therefore consider including trypsin inhibitors, as they are used during pancreatic slice cultures obtained from healthy donors [10]. Additionally, the removal of tumors from their native environments causes disruption to the vascular and lymphatic system. While cells within slices may use the local vasculature for motility and migration ex vivo, slice cultures are cut from the systemic circulation. As such, the ability to study biology of the TME that is regulated by communication with cellular elements and soluble factors derived from the vasculature and neighboring tissues are limited.

Another limitation of the slice culture model is that the donor tissue availability is a limited biological resource. Research efforts should focus on aspects of tumor biology that are not well recapitulated by alternative in vivo and in vitro models. Moreover, the study of metastatic PDAC is more challenging as typically only small amounts of tissue are available, given that patients with advanced disease are not typically eligible for surgical resection. In all cases where tumor tissues are obtained following surgical resection, the resected tumor is initially handled by a pathologist to mark resection margins for subsequent analyses. Therefore, it is not currently feasible to study the invading edge of a tumor. Thus, most research tissues are located near the tumor core, or a visually discernable region from the tumor margin.

Lastly, cellular heterogeneity within tumors remains a challenge for experimental planning, data analysis and interpretation. To account for tumor heterogeneity, sequentially cut slices should be matched from similar tumor regions for drug testing. While this is laborious for researchers, heterogeneity normalization using matched slices is essential for downstream analyses of cellular and biochemical assays. While intra-tumoral heterogeneity remains a formidable challenge, normalizing the data between different patients requires careful attention. In many cases tumors from patients are removed in preoperative therapy (chemotherapy and/or radiation). These treatments impact tumor cell viability, and thus it is essential to assess the slice viability prior to experimental manipulation in all cases. 

While pancreatic slices from healthy (non-tumor) donors were analyzed for up 21 days [15], longer term studies (weeks to months) have not been performed on slices from human (or murine) PDAC specimens. These studies are most likely feasible since human and animal models slices from tumors of the brain and lungs are shown to survive in long-term cultures and will allow for the in-depth molecular investigation of human tissues using viral vectors [53,54] Future studies using slices should continue to evaluate the viability and feasibility of longer term slice culture and its impact on tumor biology.

## 6. Conclusions

The use of ex vivo tissue is proven to be a valuable model for clinical and basic research studies. While drug modeling in slice cultures may accurately predict an in vivo response, the 3D organoids derived from patients are a renewable and scalable resource. As such, they are a more feasible tool for drug sensitivity, synergy, and resistance testing. However, it should be noted that the drug testing in slice cultures might be more appropriate when an investigational drug is dependent on multiple aspects of the pancreatic tumor microenvironment (such as the immune cells or ECM composition) since 3D organoids only contain epithelial cells. Further research with organotypic slices is needed to better understand these paracrine signaling interactions between cellular and the extracellular environment. To this extent, generating spatial information will be informative in the future to decode paracrine signaling mechanisms within PDAC. Understanding how pancreatic cancer cells communicate with their microenvironments will be essential to provide patients with a durable treatment for this devastating disease. Ultimately, the fidelity of all models to predict the responses and outcomes in patients requires validation in well-controlled clinical trials.

## Figures and Tables

**Figure 1 cancers-13-04991-f001:**
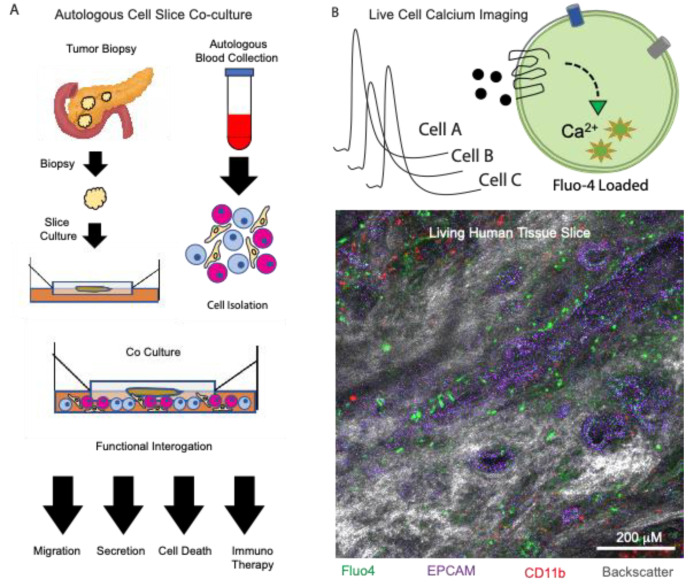
Using organotypic slices to examine paracrine signaling in the tumor microenvironment. (**A**) Processing of tissue slices from donor biopsies while performing autologous cell isolation and co-cultures. Slices are analyzed downstream using functional assays. (**B**) Live cell calcium imaging using an ex vivo tissue slice from a human pancreatic donor. In situ cytolabeling was performed using the calcium indicator dye fluo4 (green), epithelial cell marker epcam (purple), immune cell marker CD11b (red), and the tissue backscatter (gray).

**Figure 2 cancers-13-04991-f002:**
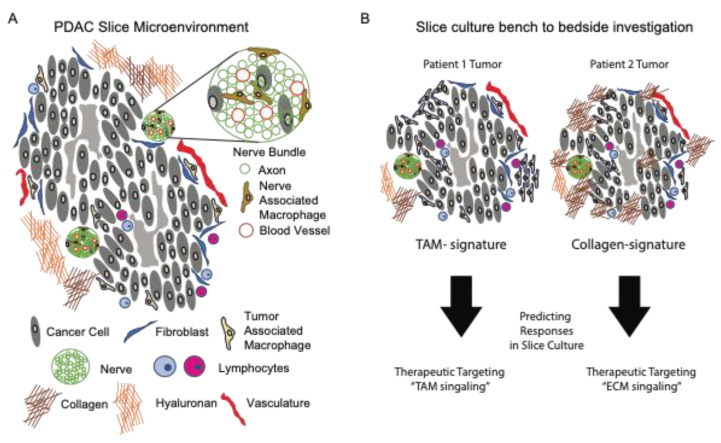
Personalized medicine approach using ex vivo preparations of living pancreatic tissue slices. (**A**) The tumor microenvironment of living slices from PDAC tissues is comprised of both cellular and a-cellular (extracellular matrix) compartments. (**B**) Tumor slices from different donors have distinct features that contribute to tumor killing efficacy of therapeutics. These features, such as tumors with highly enriched tumor-associated macrophages (left panel) or dense collagen deposition (right panel) will guide therapeutic testing of unique microenvironmental signatures.

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
