# Peer review of "Using Organotypic Tissue Slices to Investigate the Microenvironment of Pancreatic Cancer: Pharmacotyping and Beyond"

_cancers, 2021, doi:10.3390/cancers13194991_

Round 1

Reviewer 1 Report

The article review the importance of Organotypic slices over regular Organoids. Overall this review does a good job to do this. Please find below a few comments to further improve the manuscript.

Comments:

  • Please check punctuation and commas throughout the manuscript (e.g., see abstract)
  • Page 3 discusses different approaches to study cellular and multicellular interactions in the pancreatic TME, but 3.1 and 3.2 are same approaches (organoids)
  • Page 5, line 197 moves from one research group to the other, but it is very generic “The authors observed…” – It doesn’t state which authors, is it still the previous team or a new research group? It looks like their own research but it is a bit confusing.
  • Lastly, in the keywords they mention organoids/hydrogels. But within the review, they only touch on the keywords in two paragraphs to introduce them. There is no direct comparison between organoids vs organotypic slices specifically, so I do not see the need to include it in the keywords. The paper focuses entirely on the rationale of using organotypic slices, so I would keep the keywords concise.

Author Response

“Please check punctuation and commas throughout the manuscript (e.g., see abstract).”

The manuscript has been checked over again thoroughly for errors in puncation.

“Page 3 discusses different approaches to study cellular and multicellular interactions in the pancreatic TME, but 3.1 and 3.2 are same approaches (organoids).”

We thank the reviewer for pointing out this similarity. On page 3 we have decided to merge the sections 3.1 and 3.2 into unified organoid focused section. As such the new 3.1 discusses cellular and multicellular interactions on the TME using alternative models to ex-vivo slices (i.e., 3D organoid models).

“Page 5, line 197 moves from one research group to the other, but it is very generic “The authors observed…” – It doesn’t state which authors, is it still the previous team or a new research group? It looks like their own research but it is a bit confusing.”

We apologize for any confusion in the readibility of the manuscript. The reviewer mentioned line 197 has now been clarified accordingly to the new lines 215-220 where we clarify that the authors are referring to Seo et al.,.

“Lastly, in the keywords they mention organoids/hydrogels. But within the review, they only touch on the keywords in two paragraphs to introduce them. There is no direct comparison between organoids vs organotypic slices specifically, so I do not see the need to include it in the keywords. The paper focuses entirely on the rationale of using organotypic slices, so I would keep the keywords concise.”

We concure with the reviwer and we have decided to remove “organoid” from the keywords section of the manuscript (see line 32).

Reviewer 2 Report

The manuscript by Weitz et al reviewed pancreatic tissue slices as a model system that recapitulates many aspects of the tumor microenvironment. This manuscript has been very well written, but the following points could be addressed.

Comments:
1.  More detailed information regarding the procedures to prepare pancreatic tissue slices from patient tumors and maintain them would be helpful.

2. The advantages of pancreatic tissue slices have been extensively described, but their limitations have not been fully discussed. The authors could more extensively discuss the limitations and future research directions of this approach.

Author Response

“More detailed information regarding the procedures to prepare pancreatic tissue slices from patient tumors and maintain them would be helpful.”

We thank the reviwer for the opportunity to address some of the details regarding the procedure. We have included a detailed paragraph addressing the preparation procedures of slices. Please see the new lines 153-167, which provides additional information on slice preparation, tissue handling, and culture.

“The advantages of pancreatic tissue slices have been extensively described, but their limitations have not been fully discussed. The authors could more extensively discuss the limitations and future research directions of this approach.”

We agree that additional information regarding the limitations and future research prospective will help guide readers with this review. As such we have included additional limitations of the human slice model regarding metasatic disease and the invading tumor edge lines 327-334. In addition, future directions are proposed lines 345-351.

Reviewer 3 Report

This is a comprehensive assay on the advantages of employing organotypic tissue slices to investigate pancreatic cancer microenvironment, which becomes highly suitable and at the same time will surely benefit the especial issue on “Tumor Microenvironment of Pancreatic Cancer”. Indeed, comparing the pros and cons of using pancreatic tissue slices rather than other well established 3D models is inspirational and will surely drive future investigations targeting the TME for cancer therapy.

Author Response

Reviewer 3.

No additional revisions were suggested.
